# Sustainable development goals and multisectoral collaborations for child health in Cambodia: a qualitative interview study with key child health stakeholders

Daniel Helldén [1], Serey Sok,[2] Thy Chea,[3] Helena Nordenstedt [4], Shyama Kuruvilla,[5] Helle Mölsted Alvesson [1], Tobias Alfvén[6,7]

[1]Department of Global Public Health, Karolinska Institutet, Stockholm, Sweden
[2]Research Office, Royal University of Phnom Penh, Phnom Penh, Cambodia
[3]Malaria Consortium, Phnom Penh, Cambodia
[4]Department of Global Public Health, Karolinska Institute, Stockholm, Sweden
[5]UHC and Live Course Division, WHO, Geneva, Switzerland
[6]Global Public Health, Karolinska Institute, Stockholm, Sweden
[7]Sachs' Children and Youth Hospital, Stockholm, Sweden

**Correspondence to**
Dr Daniel Helldén;
daniel.hellden@ki.se

## ABSTRACT

**Objectives** Multisectoral collaboration highlighted as key in delivering on the Sustainable Development Goals (SDGs), but still little is known on how to move from rhetoric to action. Cambodia has made remarkable progress on child health over the last decades with multisectoral collaborations being a key success factor. However, it is not known how country stakeholders perceive child health in the context of the SDGs or multisectoral collaborations for child health in Cambodia.

**Design, settings and participants** Through purposive sampling, we conducted semistructured interviews with 29 key child health stakeholders from a range of government and non-governmental organisations in Cambodia. Guided by framework analysis, themes, subthemes and categories were derived.

**Results** We found that the adoption of the SDGs led to increased possibility for action and higher ambitions for child health in Cambodia, while simultaneously establishing child health as a multisectoral issue among key child stakeholders. There seems to be a discrepancy between the desired step-by-step theory of conducting multisectoral collaboration and the real-world complexities including funding and power dynamics that heavily influence the process of collaboration. Identified success factors for multisectoral collaborations included having clear responsibilities, leadership from all and trust among stakeholders while the major obstacle found was lack of sustainable funding.

**Conclusion** The findings from this in-depth multistakeholder study can inform policy-makers and practitioners in other countries on the theoretical and practical process as well as influencing aspects that shape multisectoral collaborations in general and for child health specifically. This is vital if multisectoral collaborations are to be successfully leveraged to accelerate the work towards achieving better child health in the era of the SDGs.

## STRENGTHS AND LIMITATIONS OF THIS STUDY

⇒ Using semistructured interviews, diverse themes around the complex phenomenon of multisectoral collaboration for child health could be explored to reach high information power.
⇒ The study included a relatively large sample of child health stakeholders at a national level with unique insights into multisectoral collaboration and knowledge of the Cambodian context.
⇒ The sample participants interviewed is unbalanced in terms of gender and expertise in different SDG areas.

## INTRODUCTION

Halfway until the United Nations Sustainable Development Goals (SDGs) are to be achieved, practitioners, experts and policy-makers are trying to speed up the pace of progress on child health. This has become even more urgent with the setback of the COVID-19 pandemic which left 147 million children out of proper education, rising child labour and significantly higher rates of malnutrition and over 22 million children missing essential vaccinations.[1 2] Over the last decades, it has become evident that progress made in other sectors heavily impact the possibility to make progress on child health and well-being.[3 4] Child survival is included in SDG 3 (good health and well-being) while the broader aspects of child health and well-being is captured by many different SDGs, for instance, SDG 2 (zero hunger), SDG 4 (quality education) and SDG 5 (gender equality). Further, progress on child health and well-being is essential for tackling poverty and promote the development of societies.[5] Moving beyond mere child survival, there is now a larger focus on enabling children to thrive and reach their full potential.[5 6]

Multisectoral collaborations have long been seen as critical for achieving gains in health and well-being when it comes to

BMJ

universal health coverage, non-communicable diseases and succeeding in governing multisectoral issues going back to the WHO Constitution and the Alma Ata Declaration.[7–9] For child health a multisectoral approach to areas such as nutrition[10] and education[11 12] have been studied, however, there is lack of understanding how multisectoral collaborations work out on a country level. Further, a generic analysis of the linkages between the SDGs and child health found that there are many synergies between making progress on the SDGs and accelerating progress on child health, suggesting that multisectoral collaboration could harness synergies and better handle trade-offs between the SDGs and child health.[13]

During the Millennium Development Goal era, many countries made significant gains in child health, and approximately half of the reduction in child mortality between 1990 and 2010 have been attributed to investments in sectors outside of health.[14] Cambodia was one of the fast-track countries and made significant progress including succeeded in lowering the under-5 mortality from 116 to 29 deaths per 1000 live births from 1990 to 2015.[15] Many challenges persist, however, with significant inequalities between rural and urban areas, lower than desired educational attainment and sub-optimal water and sanitation conditions in schools and residential areas.[16] It has been shown that multisectoral efforts, such as the ID Poor programme, have been successful in reducing poverty and collaborative initiatives between non-health sectors have become a cornerstone of the maternal and child health strategy in Cambodia.[17–19]

Cambodia has managed to improve the health and well-being of children over a short period of time while using collaborations across sectors to do so among other changes. However, it is not known how child health stakeholders have been influenced by the SDGs or how they theorise multisectoral collaborations, here defined as 'multiple sectors and stakeholder intentionally coming together and collaborating in a managed process to achieve shared outcomes and common goals',[20] versus the actual practice of conducting such collaborations. This knowledge could inform current and future multisectoral collaborations on critical theories and key success factors and obstacles when initiating and implementing such a collaboration. Hence, our aim was to understand how stakeholders in Cambodia perceive the SDGs, child health in the era of the SDGs and multisectoral collaborations for child health in Cambodia.

## METHODS
### Study design and setting
Guided by the The COnsolidated criteria for REporting Qualitative (COREQ) recommendations[21] and the concept of information power,[22] this study uses semistructured interviews to investigate how Cambodian stakeholders perceive the SDGs, the concept of child health in the era of the SDGs and multisectoral collaborations for child health in Cambodia. The country is governed primarily through the national government, which consists of the council of ministers led by the prime minister while the parliament (national assembly and senate) have legislative power. Administratively, the country is divided into provinces, districts, communes and villages.[23] During the last decade, the government has incrementally favoured a more decentralised approach where districts and commune government officials are given more funding and implementing power.[24] Collaboration between government and non-government stakeholders primarily occur on two levels, the national or subnational (district or commune) level and been characterised by an increased role of the government in leading and coordinating collaborations.[25 26] The Ministry of Health and its National Maternal and Child Health Centre is responsible for health services throughout Cambodia, often working in committees or technical groups with other relevant ministries and in collaboration with international and Cambodian non-governmental organisations. At the subnational government level, provincial health departments and operational health districts lead the implementation of national strategies and technical guidelines together with national and local non-governmental organisations in a more ad hoc fashion.

### Participant identification and recruitment
Key child health stakeholders with country-specific knowledge as well as non-health sector stakeholders on a national level in Cambodia were identified for participation by the research team. Participants were purposively selected based on predefined criteria of having expertise in child health or being from a non-health sector (eg, water and sanitation, agriculture, infrastructure) but with implementation knowledge of how child health interacts with other sectors in Cambodia. Efforts were made to recruit participants from many different sectors, including having participants from inside and outside of government. Further, the recruitment of participants was aimed to be balanced in terms of sex and seniority. The outreach to participants was done by DH, SS and TC through email and phone. The expected total number of participants was 30, balancing the need for reaching satisfactory information power[22] and feasibility.

### Data collection
A total of 29 participants were interviewed between April and June 2020. Information was given verbally to all participants on the purpose of the study, what their involvement in the study would be, the risks and benefits of taking part in the study, and that they had the right to decline participation or withdraw from the study at any time for any reason. Participants were asked to sign an informed consent form, written in Khmer before the interview started. The interviews were held in Khmer by authors SS and TC, audiorecorded and transcribed verbatim into English. The interviews took place in Phnom Penh city vicinity, at the participant's place of employment or other convenient but private location for

the participant. An interview guide was developed based on established multisectoral frameworks; the SDG synergies framework,[27] health in all policies approach[28] and multisectoral collaborative model presented by Kuruvilla *et al*[20] (see online supplemental material 1 for interview guide). The interview started with general background information on the participant, including the work experience in different sectors as represented by the Cambodian SDGs and moved on to the perception of the SDGs, child health and multisectoral collaboration and then focused on multisectoral collaboration for child health within the Cambodia context (identification of problem, design, implementation and monitoring of the collaboration as well as relationships and capacity building activities). All types of collaborations between at least two or more sectors that had the explicit goal in some way to improve child health were considered during the interview. Two pilot interviews were held where after the interview guide was slightly adjusted for clarity.

## Data analysis

Transcripts were imported into NVivo software for analysis. The transcripts were first analysed by framework method analysis[29] by which the transcripts were read in full by DH, then coded through identification of meaning units, combining these into subcategories and then grouped into overarching categories and lastly themes following the standard methodology. The themes, categories and subcategories were inductively developed without prior anticipations[30] and continuously developed during the review of the transcripts. As such, the concepts of child health, SDGs and multisectoral collaboration emerged inductively. The coding was cross-checked by HMA and the analysis was continuously discussed with SS and TC to improve trustworthiness and validity.[22]

## Patient and public involvement

No patients or public representatives were directly involved in the design, conduct or reporting of this study. The findings will be disseminated and discussed with involved stakeholders. A reflexivity statement can be found in online supplemental material 2.

## RESULTS

A diverse set of perspectives were provided by the participants (see table 1 for participant characteristics) on the research questions. Out of these, two main themese merged in addition to several subthemes and categories (table 2, see online supplemental material 1 for full coding tables and COREQ checklist). The first theme related to the views of the participants on how the SDGs and expanded view on child health enable change the and the second main theme detailed the gap between theory and real-world complexities of conducting multisectoral collaborations for child health.

## SDGs and expanded view on child health enable change

Overall, interviewees reflected on the willingness by the national government to adopt the SDGs, how the possibility to achieving the SDGs depends on the outlook for the country while concluding that child health is a multisectoral topic at heart and that with the introduction of the SDGs the participants had set higher ambitions for child health and well-being.

### Possibility for action due to SDGs

The 2030 Agenda and the SDGs were thought of as a universally relevant vision for sustainable development, providing a concrete roadmap or guide for each country. Comparing with the previous Millennium Development Goals, participants reflected on how the SDGs represent a more complex and detailed set of objectives that mirror actual conditions in the country. There was an overall agreement that the SDGs showcase that health is a multisectoral issue more clearly than during the Millennium Development Goals era. However, although the commitment to and leadership of the national government of Cambodia in adopting and implementing the Cambodian SDGs were evident, some participants noted the discrepancy between the highly set ambitions of the contextualised SDGs with the resources and work committed.

> That's the difference in perspectives between policymakers and implementers. The implementers in the ministry will complain about having lots of challenges and risks which could lead to a lower result. So, the plan to achieve many things by 2030 has already been written down. However, the implementation need budget and solutions to the challenge.—Nr 21, nongovernmental organisation

### Higher ambitions for child health, a multisectoral area at heart

Focusing on child health, most regarded children as people under the age of 18 and emphasised that physical and mental health are of equal importance to children. Interviewees detailed a range of linkages between child health and other sectors, mostly focusing on education and schooling, nutrition and other general societal conditions such as physical safety, environment, economic development and social protection systems. Overall, there was a strong notion of indivisibility between child health and its determinants, making the case that child health by definition is a multisectoral issue with all sectors responsible for its improvement.

> Like I mentioned, child health consists of physical, mental and social health. So, we need all relevant institutions to improve physical, mental and social health. We can't miss anyone to work on it.—Nr 5, governmental organisation

Interviewees put an emphasis on the family as responsible for the child's health, while other stakeholders (government, international organisations and private sector) play an important role in shaping the determinants of child health in Cambodia. They further urged a concrete focus on preventive measures, improving quality and reach of health services related to the child and the

**Table 1** Participant characteristics

| No | Sex | Years worked | Organisation | Work sector experience according to Cambodian Sustainable Development Goals |
|----|-----|--------------|--------------|-----------------------------------------------------------------------------|
| 1 | Male | 6–14 | Governmental | 7, 13, 14, 15 |
| 2 | Female | >15 | Governmental | 3, 5, 6 |
| 3 | Female | >15 | Governmental | 5, 17 |
| 4 | Male | >15 | Governmental | 1, 3 |
| 5 | Male | >15 | Governmental | 3, 4, 17 |
| 6 | Female | 1–5 | Governmental | 3, 16 |
| 7 | Male | >15 | Governmental | 3, 4 |
| 8 | Male | >15 | Governmental | 1, 2, 3, 4, 5, 6, 7, 10, 11, 12, 16, 17 |
| 9 | Male | >15 | Governmental | 3, 4, 5, 16 |
| 10 | Male | >15 | Governmental | 4, 17, 18 |
| 11 | Male | >15 | Governmental | 4, 17, 18 |
| 12 | Male | 6–14 | Governmental | 1, 2, 3, 6, 16 |
| 13 | Female | 6–14 | Governmental | 1, 3, 17 |
| 14 | Male | 6–14 | Governmental | 17, 18 |
| 15 | Male | 6–14 | Non-governmental | 2, 3, 4, 6, 13 |
| 16 | Male | >15 | Non-governmental | 2, 3 |
| 17 | Male | >15 | Non-governmental | 5, 10, 16, 17 |
| 18 | Male | 6–14 | Non-governmental | 2, 3 |
| 19 | Female | 1–5 | Non-governmental | 1, 3, 4 |
| 20 | Male | >15 | Non-governmental | 2, 4, 8 |
| 21 | Male | >15 | Non-governmental | 16, 17 |
| 22 | Female | >15 | Non-governmental | 2, 3, 4, 6 |
| 23 | Male | >15 | Non-governmental | 3, 4, 5, 6 |
| 24 | Female | >15 | Non-governmental | 2, 3, 5, 6 |
| 25 | Male | >15 | Non-governmental | 1,2,3, 4, 6 |
| 26 | Male | >15 | Non-governmental | 2, 3 |
| 27 | Male | 6–14 | International | 2, 3, 17 |
| 28 | Female | 6–14 | International | 1, 2, 3, 16, 17 |
| 29 | Male | >15 | International | 2, 3, 4, 6, 17 |

Cambodia SDG (1) no poverty, (2) zero hunger, (3) child health, (4) quality education, (5) gender equality, (6) clean water and sanitation, (7) affordable and clean energy, (8) decent work and economic growth, (9) industry, innovation and infrastructure, (10) reduced inequalities, (11) sustainable cities and communities, (12) responsible consumption and production, (13) climate change, (14) life below water, (15) life on land, (16) peace, justice and strong institutions, (17) partnership for the goals and (18) mine/ERW free.

family to improve child health further. Lastly, interviewees made the case for a life course approach to child health and setting a higher ambition for children with a focus on child growth and stronger acknowledgement of the rights of the child.

> To understand about the needs of children, we need to understand the growth of them first. Children's development consists of children before birth, children after birth to two years old, children in kindergarten and primary school, and children in high school. The development of children on physical health,

education and morality are ongoing process.—Nr 5, governmental organisation

### Gap between theory and real-world complexities
When discussing multisectoral collaborations for child health, it became clear that there is a step-by-step linear process of thinking around the collaboration and its activities while aspects influencing the collaboration shape and direct the process in non-linear fashions. Participants also critically assess the collaborations, identifying success

**Table 2** Main themes, subthemes and categories

| Themes | SDGs and expanded view on child health enable change | | Gap between theory and real-world complexities | | |
|---|---|---|---|---|---|
| Subthemes | Possibility for action due to SDGs | Higher ambitions for child health, a multisectoral area at heart | Planned linear process of collaboration | Real-world complexities shaping the collaboration | Critically assessing collaboration |
| Categories | SDGs provide a common vision and guide Government commitment to and leadership of SDGs Discrepancy between ambition and actual work | Definition of child health Child health linkages across sectors Aspects of the health system and actors unique to children Special considerations for children | Identifying and framing problem Actors and topics Planning Coordination Implementation Monitoring and evaluation Dissemination | Funding Relationships Enabling environment Capacity building | Success factors Obstacles |

SDG, Sustainable Development Goal.

factors and obstacles for these types of collaborations in Cambodia.

### Planned linear process of collaboration

The beginning of a multisectoral collaboration typically began with the identification and framing of a problem. This could be from a top-down approach, whereby government ministries identified a gap or need, or through policy or development plans while funding opportunities and the own organisational strategy or values could be other ways of identifying a problem. On the other hand, interviewees also described a bottom-up approach of problems being identified through routine data or findings from the grassroot level, complemented by listening and learning from community or subnational stakeholders. The identified problem was often not primarily concerned with health but noted that child health might stand to benefit as an effect of a successful solution to the problem. The problem was typically framed in a detailed problem statement following involvement of many stakeholders in the collective process, often using research in some way to narrow the problem.

So, the needs can be identified through annual reports and through our observation in different sectors. Sometimes, we also do things following the donors' research and findings.—Nr 1, governmental organisation.

They (government officials) collected all data from institutions under Ministry of Health. Then, they identified the challenge and priority action plans for next year. Besides, each unit need to monitor their annual results and to identify the priority action plans. That's how the Ministry of Health and different units identify the needs on child health, status, results and ways forward to reach SDGs.—Nr 6, governmental organisation.

The stakeholders involved in the discussed collaborations varied substantially, however, the government (at national or subnational level) was seen as a natural leader of collaborations while non-governmental organisations often organised in networks. Interviewees expressed territory feelings, with relatively strict boundaries between stakeholders and a critical view of government by the non-governmental organisations and vice versa.

I am not blaming the government institutions, but there are some institutions which have too clear boundaries on their responsibilities and work. This leads to failure in our work.—Nr 29, international organisation.

Planning of the collaboration was seen as a complex, detailed and resource demanding process. Often not formalised, a capacity assessment of the stakeholders in the collaboration, primarily focusing on implementation capacity and not on specific knowledge or expertise in a particular sector or area, was usually done at this stage, with the division of activities based on this assessment. If there was not enough implementation capacity to solve the problem identified, the collaboration could not begin. During the planning process interviewees noted that prioritisation of activities was done depending on the funding requirements and to secure buy-in from certain stakeholders (particularly national government) seen as necessary for the success of the collaboration.

For example, they (government servants) may plan 20 activities, but receive inadequate budget. So, they prioritize the activities to be done. According to my observation, district level is the same. They engage politics into their work. They like infrastructure development more than social development because it is eye-catching and visible.—Nr 9, governmental organisation.

Coordination was done in various ways depending on the collaboration; however, there were usually a common information sharing mechanisms, focal points at each stakeholder or joint committees with continuous coordination often built on somewhat already existing structures. There was also a clear division of responsibilities,

although participation in joint coordination could be difficult to achieve and often those who coordinate do not have decision-making power. Clear leadership of the collaboration was seen as paramount, with coordination succeeding or faltering based on the competence and willingness of the leader. As such, coordination was both a formal and informal process. Indeed, power and hierarchies shaped the coordination efforts where power imbalances or competition for funds between stakeholders could threaten the whole collaboration.

> Those people also need to have the authorization in decision-making in the meeting. In the past, there were people who attended the meeting, but did not do what were discussed. It was useless when people came to listen, but didn't share to their management and colleagues.—Nr 15, non-governmental organisation.

Implementation of the collaboration tried to follow the planning and set coordination mechanisms. However, collaborations were able to change depending on a change in the context or influencing aspects such as the COVID-19 pandemic or funding changes. Interviewees emphasised the difference between the national and subnational level in terms of the collaboration, with larger collaborations having an administrative or policy function at the national level while implementation occurred at the subnational level. This structure often led to increased complexities, with a different set of stakeholders needing to be involved at the different levels and the subnational system having its own set of priorities.

> National level only work on policy. So, implementation goes to community level. I think that we should focus on provincial and communal level first to let them implement the work. We should also try to integrate the coordination with national level too by using forum to meet and discuss on the challenge.—Nr 28, international organisation.

Monitoring and evaluation were seen as integral to the collaboration, enabling learning and improvement of the collaboration itself and its activities and serving as the main accountability mechanism. The responsibility of conducting the monitoring and evaluation varied depending on the context and funding available, with external evaluation being seen as favourable if it could be funded. The national government and international organisations relied heavily on monitoring and evaluation for making decisions about the collaborations. However, it was seen as hard to move beyond pure outputs, with quantitative indicators believed to be most reliable, and to attribute successes or failures to different stakeholders in the collaboration.

Dissemination of the collaboration and its activities were primarily thought of as information spreading, trying to raise awareness of the identified problem and engage the public and relevant stakeholders at national and international level in the efforts to solve it. It was also

deemed important as a means of ensuring recognition from national level government ministries or the international community for the work done.

> We shared a lot, especially early childhood development program. We shared at provincial level and national committee on children education. We invited those committees to see our target location and our work. So, we disseminated a lot. Nr 22, non-governmental organisation.

### Real-world complexities shaping the collaboration

There were a range of aspects influencing the process, often challenging the idea of a step-by-step linear approach of the collaboration. The most prominent aspect throughout was the funding, interviewees described the budget as the greatest limitation to the collaboration and called for more governmental funding at the national and subnational level for multisectoral collaborations. Funding was seen as the most important source of power in the collaboration. Leadership roles, agenda setting and decision-making were mostly done by the organisation that controlled the funding.

> More importantly, we need the money to be available at sub-national level. The partners are all institutions. If the government can't manage to work on everything, we can ask civil society to help working on that. Nowadays, we are sceptical with non-governemental organisations. But, we also have example of government providing budget for non-governemental organisations to work. Nr 4, governmental organisation.

Relationships between the collaborators could facilitate or hamper the collaboration, with tensions between non-governmental organisations and the government existingand at the same time conflicts between government ministries or civil society networks that added complexity. For this reason, many collaborations tried to actively build relationships over time particularly between coordination focal points or joint committees, seeing mutual understanding leading to trust and confidence in the collaboration.

> The collaborative work also became better. During my time at education sectors, the relationship between partners was going very well, and we were happy to share any documents or data.—Nr 23, non-governmental organisation.

> There are many non-governemental organisations working to promote children. The government don't even know who they are. Some non-governemental organisations don't care about networking with the government too. This is the challenge according to my observation as a person in the middle of the two institutions. Both have their own weakness. Some non-governmental organisations do not know what the ministry have. For example, some non-governemental organisations do not know about existing guideline,

plan or projects to work consistently. They only focus on their own work, and not pay attention to what others do to work collaboratively on the topic.—Nr 2, governmental organisation.

Capacity building was deemed to be key for the sustainability of the collaboration and its activities, particularly at the subnational or implementation level, although demanding significant resources and the actual method varied depending on the type of collaborations and the stakeholders involved.

Whenever there are requests from anyone or any organisations, we always respond and provide the training or sharing of experiences. We never hide our knowledge. We don't even charge them. We do it from our heart and soul.—Nr 10, governmental organisation.

An enabling environment, particularly concerning policy and national governmental direction within which the collaboration took place, was seen as being of crucial importance. The introduction and adoption of the 2030 Agenda and the Cambodia SDGs, subnational plans for development and national level plans promoted the idea of multisectoral collaboration. Government ministries that actively promoted or worked in multisectoral ways or through multisectoral committees, although not always successful, further promoted the advantages of tackling problems in a multisectoral fashion.

The main thing is whether or not they have the commitment to work together. When commitment on that occur, the work can be done easily because visions created in country and global level has already been created.—Nr 17, non-governmental organisation.

### Critically assessing collaborations

Interviewees reflected critically on their collaborations and had through experience identified some key success factors and often faced obstacles of multisectoral collaborations in Cambodia. Having clear responsibilities with agreement on division of activities, leadership from all and functioning monitoring and evaluation as well as a common vision and understanding based on continuous learning in an open environment where benefits and goals were explicit seem to be key success factors. Further, many emphasised the necessity of securing buy in, trust and commitment from all stakeholders in the collaboration from the beginning with the national government having a special role in all collaborations.

Problems always occur. To work well with each other, we need to have collaborative plan with everyone's ownership. Secondly, we need to build trust and not allow any mistrust to happen.—Nr 27, international organisation.

We also work closely and indirectly with selected institutions which have the most power.—Nr 3, governmental organisation.

Obstacles identified were lack of funding or long-term sustainability of the collaboration and its activities, with politics on subnational and national level could mean unfavourable conditions for a collaboration or simply competing priorities or work of the stakeholders in the collaboration. There could also be a sense of a lack of accountability towards each other or the thought beneficiaries, with sometime faltering commitment to work together, lack of transparency of funds or efforts and difficulty of attributing failures or successes.

For instance if we are looking among 25 sub national civil society working group at the provincial/municipal level, there was only 50% who were active. Among these half, only 20 to 30 % who were very active in fulfilment of their collaborative work.—Nr 20, non-governmental organisation.

## DISCUSSION

In this study, we found that the adoption of the SDGs led to an increased perceived possibility for action and higher ambitions for child health, perpetuating child health as a multisectoral issue. Further, there seems to be a gap between the desired step-by-step theory of conducting multisectoral collaboration and the real-world complexities of conducting such collaborations for child health in Cambodia. This is the first study to provide in-country insights that can be transferable on multisectoral collaborations for child health, overcoming some of the key methodological gaps noted by Glandon *et al*[31] including describing power dynamics, type of governance arrangements and a diversity of stakeholder experiences.

The expanded view of child health and higher ambition for children to thrive led to a more compelling case for multisectoral collaborations to have a collaborative advantage over single-sector or single-stakeholder efforts. The 2030 Agenda and the SDGs influence social norms at a global, country, organisational and individual level.[32] The widespread knowledge of the overarching ambition and content of the SDGs in our study serve to exemplify the notion of universality of the 2030 Agenda, and the normative significance of universality in a country context.[33] Further, the perceived high ambition of the SDGs, the diversity of topics covered in the SDGs and their interlinked nature might shift norms to be more favourable towards multisectoral collaboration, in line with Huxham's theory of collaboration advantage.[34] Placing children firmly in the centre of the SDGs in Cambodia might also allow for a revitalisation of action and enable policy-makers and practitioners to use the interlinkages within the SDGs to build multisectoral collaboration for child health.[35 36]

Multisectoral collaborations depicted by the participants in this study showcase that there is often no linear process but rather ongoing non-linear flow of activities that intentionally lead to a multisectoral collaboration (see online supplemental material 1 for illustrative examples

of multisectoral collaborations). The rational logic of inquiry theory whereby one step leads to the next one until a decision is made and action is implemented and evaluated originally proposed by Dewey[37] were perceived by the participants to be the desired theory or process of collaboration. However, as showcased by Kuruvilla and Dorstewitz[38] previously, the collaborations described somewhat mimic the multisectoral collaboration model[20] which rests on dynamic networks and changing contexts. There was usually a capacity assessment of the potential or included stakeholders at the beginning of the collaboration; however, it was usually described as informal or focused on securing funding and political buy-in rather than ensuring the implementation capacity of the collaboration, which could be why many collaborations had to divert from the desired linear process. Indeed, in our study participants singled out funding as an enabler and obstacle as well as a significant source of power in multisectoral collaborations. As noted by Rasanathan *et al*,[39] if multisectoral collaborations for health are to succeed appropriate financing systems that incentivise these collaborations must be in place, and the multisectoral monitoring and evaluation mechanisms allow for accountability. Conflicting perspectives between stakeholders, particularly government and non-governmental stakeholders, has been documented in Cambodia[25 26 40 41] and in other settings.[42 43] In our study, there was a difference between interviewees from governmental organisations versus those from non-governmental particularly concerning the commitment and ability of the government to support and participate in multisectoral collaborations for child health. Although exploring this potential conflict was not the aim of this study, the emphasis of the participants on explicit and implicit territory feelings, hierarchies and power dynamics at a national and subnational level in Cambodia strengthen the need to include these concepts in collaborative theory and when designing multisectoral collaborations.[44 45]

Our limitations include that the purposive sampling led to selection bias in the recruitment of participants. As illustrated in table 1, the interviewees were slightly unbalanced in terms of gender and work experience in SDG areas. Further, although much of the implementation of multisectoral collaborations is at the subnational level the focus of this study was on the national level. Future studies might benefit from including participants with knowledge of collaborations on the subnational level. Participants were asked to reflect on one or two multisectoral collaborations to inform the answers to the questions in the interview, they might have had a positive recall bias, only including those that were successful. Given the critical assessment of the multisectoral collaborations apparent in the interviews, this seems negligible, however. Lastly, intrapersonal dynamics between the interviewer and the interviewee might affect the answers and follow-up questions. In our study, the interviews were conducted by SS and TC, both representing academic institutions and being knowledgeable of qualitative research methods

and the political landscape of organisations in Cambodia, ideally enabling both government and non-government stakeholders to express views and perceptions freely while adding credibility to the results. Although some of the findings in this study might reflect the unique Cambodia context, we believe that overall themes and conclusions are transferable to other middle-income countries and similar settings, adding valuable evidence on how stakeholders view multisectoral collaborations in general and specifically for child health. The study was designed to accomplish high information power across the five dimensions of information power,[22] however, with a broad research question and cross-case analysis the sample size was deemed to have to be relatively large to reach satisfactory information power and theoretical saturation. Information power was further increased by use of dense sampling method (purposive and specific), applied theory in the form of established frameworks for multisectoral collaborations, and high-quality dialogue in the interviews allowing for in-depth diverse multistakeholder perspectives.

## CONCLUSION

We found that stakeholders in Cambodia perceived the SDGs to inspire an expanded view on child health that enabled change and promoted multisectoral collaboration. Interviewees experienced a gap between the desired theory of conducting multisectoral collaborations for child health and the real-world complexities of engaging in such an endeavour. The findings from this in-depth study can inform policy-makers and practitioners who wish to encourage and take advantage of multisectoral collaborations for accelerating the work towards achieving better health in general and child health specifically the era of the SDGs.

**Contributors** TA, HMA and DH conceived and designed the study. TC and SS contributed to the study design and conducted the data collection. DH analysed the data together with TC, SS and HMA. DH wrote the first draft of the manuscript to which all authors (DH, TC, SS, HN, SK, HMA and TA) provided critical contributions. All authors read and approved the final manuscript. TA is the guarantor of the study.

**Funding** The work was supported by the Swedish Research Council (2018-03609).

**Disclaimer** The author is a staff member of the World Health Organization. The author alone is responsible for the views expressed in this publication and they do not necessarily represent the views, decisions or policies of the World Health Organization. The funding organisation were not involved in the manuscript's writing or the decision to submit it for publication.

**Competing interests** None declared.

**Patient and public involvement** Patients and/or the public were not involved in the design, or conduct, or reporting, or dissemination plans of this research.

**Patient consent for publication** Not applicable.

**Ethics approval** The study received ethical approval from the National Ethics Committee for Health Research in Cambodia (NECHR-023) and was exempt from ethical review from the Swedish Ethical Review Authority (Dnr 2022-00424-01). Written informed consent was obtained from all participants before inclusion in the study.

**Provenance and peer review** Not commissioned; externally peer reviewed.

**Data availability statement** No data are available. This is a qualitative study of a relatively small sample population in Cambodia. Making the dataset publicly

available could potentially breach the privacy that was promised to participants when they agreed to take part and the ethical approvals granted.

**ORCID iDs**
Daniel Helldén http://orcid.org/0000-0001-8969-2194
Helena Nordenstedt http://orcid.org/0000-0002-9226-6441
Helle Mölsted Alvesson http://orcid.org/0000-0001-6109-7203

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
