## [Reviewer comments · BMJ Open]

ARTICLE DETAILS

TITLE (PROVISIONAL)	Sustainable development goals and multisectoral collaborations for child health in Cambodia: a qualitative interview study with key child health stakeholders
AUTHORS	Helldén, Daniel; Sok, Serey; Chea, Thy; Nordenstedt, Helena; Kuruvilla, Shyama; Alvesson, Helle; Alfvén, Tobias

VERSION 1 – REVIEW

REVIEWER	Costa, Claudia University of Coimbra, Centre of Studies in Geography and Spatial Planning
REVIEW RETURNED	22-Apr-2023

GENERAL COMMENTS	The manuscript is very interesting. I just have a few comments that I hope will improve it: - line 27: for those who are unfamiliar with SDG, it would be important to refer SDG3 and 17 and the goal regarding child health- line 37: with sectors? are they hierarchical or thematic?- line 51: being the first study and only presenting this strength is not enough- line 52: if the knowledge transference is a strength, then the abstract should highlight what bet practices from cambodja can be replicated in other countries- line 75: which are the sectors that directly/indirectly work on child health? which kind of multisector collaboration is considered in the manuscript?- line 91: if It has been already shown that multisectoral efforts, have been successful in reducing poverty and collaborative initiatives between non-health sectors have become a cornerstone of the maternal and child health strategy in Cambodia, what is the novelty of this manuscript?- line 96: the relevance of this manuscript to the science is not clear. It is important to reframe introduction to clarify where the gap is and how the manuscript with contribute to decrease the gap- line 121: why only keeping with the national level? several reports state that we cannot accomplish most of the goals if we don't work at local level. therefore, considering only the national level is a limitation- line 128: why 30?- line 133: the study has been approved by a ethics committee? it is important to highlight here- line 148: experience on working on SDGs? or on knowledge regarding SDGs? or impact of their work on SDGs? it is not clear- line 148: the table should reflect the work sector- line 30: not clear
--

REVIEWER	Shikako-Thomas, Keiko McGill University
REVIEW RETURNED	13-Jun-2023

GENERAL COMMENTS	Thank you for the opportunity to review this highly relevant and timely study, relating the SDGs and child health, and offering perspectives from government and non-governmental actors. This research is much needed and can contribute to shedding light on child health in public policy agendas. The study is very well designed and the manuscript presents results in a logical and coherent manner. I offer a few suggestions to improve clarity in some aspects related to Cambodia's governance structure and clarify the conceptual use of some terms like "health", as well as the suggestion to add a few more concrete examples that can provide suggestions for solutions and moving forward in implementing SDGs towards child health in different countries. Please clarify how the "Experience according to Cambodian Sustainable Development Goals" (Table 1) was ascertained? – are these years of experience working in position, or specific to working with SDGs? Was it self identified by participants or some other objective measure? And how does that is expected to influence participants' views, or was considered in analysis? Appreciate reflexivity statements in supplemental materials It would be important to provide an overview of governance structure in Cambodia, for an international readership to be able to generalize and apply findings. In particular, what is the structure and communication among bodies responsible for child health in the country, as this surely plays an important role in some of the core themes identified (such as the gap between theory and complexity of implementation). For instance, in Results: "Overall, interviewees reflected on the willingness by the government to adopt the SDGs, how the possibility to achieving the SDGs depends on the outlook for the country while concluding that child health is a multisectoral topic at heart and that with the introduction of the SDGs the participants had set higher ambitions for child health and well-being" □ it would be important for readers to understand what is meant by "the government" (is it a the national or sub-national level, who has jurisdiction for child health and how different areas of government interact). Similarly, it would be important to present a clear vision of how authors defined "health" in the context of this project, particularly in using a framework analysis approach, to contextualize themes like "high ambitious in child health" - and if the definition of health presented in the results was inductive or deductive. Page 11 there is mention of the importance of "capacity assessment" of stakeholders – it would be important to expand on that concept, as it seems key to understanding the gap between theory and implementation – which stakeholder groups mentioned that? (is there a difference in perception of knowledge Is that in relation to SDG-specific capacity building, or child health, or implementation? □ the issue of capacity building is brought up again in Page 13, but in relation to the governmental and NGOs capacity, I wonder if these should be combined as they seem to be addressing the same theme. The topic of multisectoral collaboration is presented in Page 14, and is a very relevant one for SDGs implementation and child
---

	health. I wonder if authors had concrete examples to present in quotes (or otherwise) of how multisectoral collaborations were established towards SDGs.
--	--

VERSION 1 – AUTHOR RESPONSE

Reviewer 1	
General comment(s)	
The manuscript is very interesting. I just have a few comments that I hope will improve it.	Dear Reviewer, Thank you for your insightful comments. We have built upon your feedback and added substantial amount of secondary data and analysis. Please find our detailed responses below. Please note that the line numbers in the responses are referring to the manuscript with track-changes.
Specific comments	
line 27: for those who are unfamiliar with SDG, it would be important to refer SDG3 and 17 and the goal regarding child health	Well-noted, in the abstract we would not be able to go into detail about the SDGs due to word count limitations, however in the introduction we have added a short overview of child health and the SDGs. Please see revised introduction on lines 80-87 “Over the last decades it has become evident that progress made in other sectors heavily impact the possibility to make progress on child health and well-being.[3,4] Child survival is included in SDG 3 (Good health and well-being) while the broader aspects of child health and well-being is captured by many different SDGs, for instance SDG 2 (Zero hunger), SDG 4 (Quality education) and SDG 5 (Gender equality). Further, progress on child health and well-being are essential for tackling poverty and promote the development of societies.[5] Moving beyond mere child survival, there is now a larger focus on enabling children to thrive and reach their full potential.[5,6]”
- line 37: with sectors? are they hierarchical or thematic?	In light of the need to re-organize the abstract according to journal guidelines, please see the revised abstract which hopefully provide clarity to this question.

	Specifically, see lines 36-42 “Results: We found that the adoption of the SDGs led to increased possibility for action and higher ambitions for child health in Cambodia, while simultaneously establishing child health as a multisectoral issue among key child stakeholders. There seem to be a discrepancy between the desired step-by-step theory of conducting multisectoral collaboration and the real-world complexities including funding and power dynamics that heavily influence the process of collaboration. Identified success factors for multisectoral collaborations included having clear responsibilities, leadership from all and trust among stakeholders while the major obstacle found was lack of sustainable funding. “
- line 51: being the first study and only presenting this strength is not enough	Thank you for this comment, as per the editorial comment above the strengths and limitations bullet points should strictly refer to the methods of the study, not the results or the conclusions. Hence we have revised the bullet points to this end. See lines 52-57: “- Using semi-structured interviews, diverse themes around the complex phenomenon of multisectoral collaboration for child health could be explored to reach high information power. - The study included a relatively large sample of child health stakeholders at a national level with unique insights into multisectoral collaboration and knowledge of the Cambodian context. - The sample participants interviewed is unbalanced in terms of gender and expertise in different SDG areas. “
- line 52: if the knowledge transference is a strength, then the abstract should highlight what best practices from Cambodia can be replicated in other countries	Thank you for this remark, it is indeed important to highlight in the abstract. It is a bit difficult to outline best practices as there is a word limitation, however acknowledging and handling power dynamics and funding issues, as well as having clear responsibilities and trust among stakeholders participating in the collaboration seem to be the core

	success factors for multisectoral collaborations for child health. This is highlighted in the results in the abstract, which has been slightly revised and knowledge transference added in the conclusion section. See lines 36-47 “Results: We found that the adoption of the SDGs led to increased possibility for action and higher ambitions for child health in Cambodia, while simultaneously establishing child health as a multisectoral issue among key child stakeholders. There seem to be a discrepancy between the desired step-by-step theory of conducting multisectoral collaboration and the real-world complexities including funding and power dynamics that heavily influence the process of collaboration. Identified success factors for multisectoral collaborations included having clear responsibilities, leadership from all and trust among stakeholders while the major obstacle found was lack of sustainable funding. Conclusion: The findings from this in-depth multistakeholder study can inform policy makers and practitioners in other countries on the theoretical and practical process as well as influencing aspects that shape multisectoral collaborations in general and for child health specifically. This is vital if multisectoral collaborations are to be successfully leveraged to accelerate the work towards achieving better child health in the era of the SDGs. “
- line 75: which are the sectors that directly/indirectly work on child health? which kind of multisector collaboration is considered in the manuscript?	Thank you for the comment. In our study, we define multisectoral collaboration as Shyama et al. “multiple sectors and stakeholder intentionally coming together and collaborating in a managed process to achieve shared outcomes and common goals”. Many different sectors, if not all, can directly or indirectly work on child health. In our study, we asked participants to reflect on a collaboration between at least two sectors that had the explicit aim to

	improve child health in some way. Hence, a broad selection of collaborations were considered. We have clarified this in the method section. See lines 166-174: “The interview started with general background information on the participant, including the work experience in different sectors as represented by the Cambodian SDGs and moved on to the perception of the SDGs, child health and multisectoral collaboration and then focused on multisectoral collaboration for child health within the Cambodia context (identification of problem, design, implementation, and monitoring of the collaboration as well as relationships and capacity building activities). All types of collaborations between at least two or more sectors that had the explicit goal in some way to improve child health were considered during the interview. Two pilot interviews were held where after the interview guide was slightly adjusted for clarity.”
- line 91: if It has been already shown that multisectoral efforts, have been successful in reducing poverty and collaborative initiatives between non-health sectors have become a cornerstone of the maternal and child health strategy in Cambodia, what is the novelty of this manuscript?	Well-noted, the existence and emphasise on multisectoral collaboration for child and maternal health in Cambodia has been showcased by others previously, which we believe is important to acknowledge. However what has not been studied before is how stakeholders actually theorize or think about multisectoral collaboration for child health. For instance, have they been influenced by the SDGs? How do they start? How are they planned? What are the key success factors or obstacles for a successful collaboration? The study is the first to provide in-country insights on these topics for multisectoral collaborations for child health. We expand on the novelty of the manuscript in the beginning of the discussion section. Please see page 14 line 199-206 “In this study, we found that the adoption of the SDGs led to an increased perceived possibility for action and higher ambitions for child health, perpetuating child health as a multisectoral issue. Further, there seem to be a gap between the desired step-by-step theory of conducting multisectoral collaboration and the real-world complexities of conducting such collaborations for child health in Cambodia. This is the first study to provide in-country insights that can be transferable on multisectoral collaborations for child health, overcoming some of the key methodological gaps noted by Glandon et al.[31] including describing power dynamics, type of governance arrangements and a diversity of stakeholder experiences. “

	Please also see answer to question below.
- line 96: the relevance of this manuscript to the science is not clear. It is important to reframe introduction to clarify where the gap is and how the manuscript will contribute to decrease the gap	We have tried to provide an overview of the SDGs, the Cambodian context as well as of multisectoral collaborations in the introduction. It is indeed important to ensure that readers understand the knowledge gap that the study addresses, and we have clarified this in the end of the introduction section. Please see lines 110-121 “Cambodia has managed to improve the health and well-being of children over a short period of time while utilizing collaborations across sectors to do so among other changes. However, it is not known how child health stakeholders have been influenced by the SDGs or how they theorize multisectoral collaborations, here defined as “multiple sectors and stakeholder intentionally coming together and collaborating in a managed process to achieve shared outcomes and common goals”[20], versus the actual practice of conducting such collaborations. This knowledge could inform current and future multisectoral collaborations on critical theories and key success factors and obstacles when initiating and implementing such a collaboration. Hence, our aim was to understand how stakeholders in Cambodia perceive the SDGs, child health in the era of the SDGs and multisectoral collaborations for child health in Cambodia. “
- line 121: why only keeping with the national level? several reports state that we cannot accomplish most of the goals if we don't work at local level. therefore, considering only the national level is a limitation	We chose to focus on the national level as a first starting point of inquiry into the phenomenon and to be able to draw general theoretical learnings from the whole Cambodia context. It is indeed important to work at the sub-national or local level to bring meaningful impact. In our study, participants emphasised the difference between the national and the sub-national level particularly when it comes to implementing the collaboration. See page 11 lines 104-109 “Interviewees emphasised the difference between the national and sub-national level in terms of the collaboration, with larger collaborations having an administrative or policy function at the national level while implementation occurred

	at the sub-national level. This structure often led to increased complexities, with a different set of stakeholders needing to be involved at the different levels and the sub-national system having its own set of priorities. “ We have added the lack of focus on the sub-national level in the limitation section. See page 15 lines 251-254: “Further, although much of the implementation of multisectoral collaborations is at the sub-national level the focus of this study was on the national level. Future studies might benefit from including participants with knowledge of collaborations on the sub-national level. “
- line 128: why 30?	
- line 133: the study has been approved by a ethics committee? it is important to highlight here	The study has been approved by an ethics committee. As per the guidelines of the article this is included in the Ethics approval statement at the end of the article. See page 16 lines 300-303: “The study received ethical approval from the National Ethics Committee for Health Research in Cambodia (NECHR-023) and was exempt from ethical review from the Swedish Ethical Review Authority (Dnr 2022-00424-01). Written informed consent was obtained from all participants before inclusion in the study. “
- line 148: experience on working on SDGs? or on knowledge regarding SDGs? or impact of their work on SDGs? it is not clear	We wanted to include some overarching characteristics of the participants to allow us, reviewers and readers to assess the sample of participants included in our study. We provide information on sex, number of years worked, main work type of organisation and the work experience according to the SDGs. We choose to collect the work sector experience data according to the SDGs as we wanted to know how balanced the sample was in terms of experience working in different sectors/different SDGs. As such, we have changed the title of Table 1 to reflect this.
- line 148: the table should reflect the work sector	Please see response to the above comment.
- line 30: not clear	We have revised this sentence to make sure the objective is more clear. See line 30-31:

	“However, it is not known how country stakeholders perceive child health in the context of the SDGs or multisectoral collaborations for child health in Cambodia.”
Reviewer 2	
General comment(s)	
Thank you for the opportunity to review this highly relevant and timely study, relating the SDGs and child health, and offering perspectives from government and non-governmental actors. This research is much needed and can contribute to shedding light on child health in public policy agendas. The study is very well designed and the manuscript presents results in a logical and coherent manner.	Dear Reviewer, Thank you for taking the time to review the manuscript and providing feedback. Please see detailed responses below. Please note that the line numbers in the responses are referring to the manuscript with track-changes.
Specific comments	
Please clarify how the “Experience according to Cambodian Sustainable Development Goals” (Table 1) was ascertained? – are these years of experience working in position, or specific to working with SDGs? Was it self identified by participants or some other objective measure? And how does that is expected to influence participants’ views, or was considered in analysis?	In line with Reviewer 1 comments, we have revised the heading to reflect that it is the work experience of the participant within different sectors, reflected through the SDGs. This was self-identified by the participant (See Supplementary Material). Please see updated revised Table 1, and added statement for clarity in the methods section on lines 166-172 “The interview started with general background information on the participant, including the work experience in different sectors as represented by the Cambodian SDGs and moved on to the perception of the SDGs, child health and multisectoral collaboration and then focused on multisectoral collaboration for child health within the Cambodia context (identification of problem, design, implementation, and monitoring of the collaboration as well as relationships and capacity building activities).” We wanted to present the work experience of the participants for us, reviewers and readers to understand the sample better. As we acknowledge as a limitation, the sample is slightly unbalanced when it comes to work experience from some Cambodian SDGs (for instance SDG 9-12). The work experience was not explicitly included in the analysis, however the different perspectives between participants coming from government of non-government organisations became evident, for instance when it came to

	who was seen as the leader of the collaboration, or who should be included. See page 10 lines 69-73: “The stakeholders involved in the discussed collaborations varied substantially, however the government was seen as a natural leader of collaborations while non-governmental organisations often organised in networks. Interviewees expressed territory feelings, with relatively strict boundaries between stakeholders and a critical view of government by the non-governmental organisations and vice versa. “
Appreciate reflexivity statements in supplemental materials	Thank you, we believe this to be an important part of all research processes.
It would be important to provide an overview of governance structure in Cambodia, for an international readership to be able to generalize and apply findings. In particular, what is the structure and communication among bodies responsible for child health in the country, as this surely plays an important role in some of the core themes identified (such as the gap between theory and complexity of implementation).	Thank you for this comment, due to our efforts to balance detail with readability and keeping within the word count limit of the journal, we have kept the governance structure quite broad. In short, the Ministry of Health have the overarching lead on child health, however the actual implementation usually occurs at the sub-national level. We have provided a bit more specifics about child health see lines 136-141 “The Ministry of Health and its National Maternal and Child Health Center is responsible for health services throughout Cambodia, often working in committees or technical groups with other relevant ministries and in collaboration with international and Cambodian non-governmental organisations. At the sub-national government level, provincial health departments and operational health districts lead the implementation of national strategies and technical guidelines together with national and local non-governmental organisations in a more ad-hoc fashion.”
For instance, in Results: “Overall, interviewees reflected on the willingness by the government to adopt the SDGs, how the possibility to achieving the SDGs depends on the outlook for the country while concluding that child health is a multisectoral topic at heart and that with the introduction of the SDGs the participants had set higher ambitions for child health and well-being” it would be important for readers to understand what is meant by “the government” (is it a the national or sub-national level, who has	Thanks for the important comment, please see response to the above question. We have tried to clarify which level of governance that participants refer to throughout the manuscript. In general, it is the national government with the Ministry of Health that have led and direct efforts when it comes to child health. For instance, when it comes to identification and framing of a problem that could be solved with a multisectoral collaboration, the directive was often

jurisdiction for child health and how different areas of government interact).	top-down led by the national ministries although grass-root/sub-national identification and framing occurred. See page 10 lines 49-55 “The beginning of a multisectoral collaboration typically began with the identification and framing of a problem. This could be from a top-down approach, whereby government ministries identified a gap or need, or through policy or development plans while funding opportunities and the own organisational strategy or values could be other ways of identifying a problem. On the other hand, interviewees also described a bottom-up approach of problems being identified through routine data or findings from the grassroots level, complemented by listening and learning from community or sub-national stakeholders.”
Similarly, it would be important to present a clear vision of how authors defined “health” in the context of this project, particularly in using a framework analysis approach, to contextualize themes like “high ambitious in child health” - and if the definition of health presented in the results was inductive or deductive.	Thank you for this comment. Although definitions exist, we have tried to capture the different concepts of child health, SDGs and multisectoral collaboration through the perspectives of the participants. We have clarified the inductive approach in the method section, see lines 180-183 “The themes, categories and sub-categories were inductively developed without prior anticipations [30] and continuously developed during the review of the transcripts. As such, the concepts of child health, SDGs and multisectoral collaboration emerged inductively.” For instance, when it comes to the sub-theme “Higher ambitions for child health, a multisectoral area at heart” it also include a category on the definition of child health according to the participants. They generally though child health included people under the age of 18 and that it included both physical and mental health, while being connected to many different aspects of society. See page 9 lines 21-25 “Focusing on child health, most regarded children as people under the age of 18 and emphasised that physical and mental health are of equal importance to children.”
Page 11 there is mention of the importance of “capacity assessment” of stakeholders – it would be important to expand on that concept, as it seems key to understanding the gap between theory and implementation – which	Thank you for raising this important aspect. As we interpret the participants, an often informal assessment of capacity of many different potential stakeholders were done at the

stakeholder groups mentioned that? (is there a difference in perception of knowledge Is that in relation to SDG-specific capacity building, or child health, or implementation? the issue of capacity building is brought up again in Page 13, but in relation to the governmental and NGOs capacity, I wonder if these should be combined as they seem to be addressing the same theme.

planning stage of the collaboration in order to choose who to include and divide activities. It was primarily the possibility to implement the multisectoral collaboration activities that were assessed (usually not expertise in subject or specific knowledge) while balancing this with the need to secure buy-in from particular stakeholders (such as national or sub-national government).

When the actual multisectoral collaboration was ongoing, capacity building of the included stakeholders was seen as a key success factor to ensure the sustainability of the collaboration. But this was primarily for the ones who were already included in the multisectoral collaboration.

We see the capacity assessment at the planning stage as a somewhat separate from the focus on capacity building to ensure sustainability during the actual multisectoral collaboration, however it stands clear that recognizing the lack of capacity and acting on it is vital for a successful multisectoral collaboration.

We have elaborated on these topics in the result section, see page 10 lines 77-84

“Planning of the collaboration were seen as a complex, detailed and resource demanding process. Often not formalised, a capacity assessment of the stakeholders in the collaboration, primarily focusing on implementation capacity and not on specific knowledge or expertise in a particular sector or area were usually done at this stage, with the division of activities based on this assessment. If there was not enough implementation capacity to solve the problem identified, the collaboration could not begin.”

And included this reasoning in the discussion section, noting that this might be a reason for the gap between desired process and actual reality of the collaboration. See page 14 lines 227-231:

“There was usually a capacity assessment of the potential or included stakeholders at the beginning of the collaboration, however it was usually described as informal or focused on securing funding and political buy-in rather than ensuring the implementation capacity of the collaboration, which could be why many collaborations had to divert from the desired linear process. Indeed, in our study participants singled out funding as an enabler and

	obstacle as well as a significant source of power in multisectoral collaborations.”
The topic of multisectoral collaboration is presented in Page 14, and is a very relevant one for SDGs implementation and child health. I wonder if authors had concrete examples to present in quotes (or otherwise) of how multisectoral collaborations were established towards SDGs.	Thank you for this comment, we have added a few illustrative concrete examples of multisectoral collaboration for child health in Cambodia in Supplementary Material 1 for the interested reader. Please see Supplemental Material 1 page 21 and page 14 line 220 in the manuscript: “Multisectoral collaborations depicted by the participants in this study showcase that there is often no linear process but rather ongoing non-linear flow of activities that intentionally lead to a multisectoral collaboration (see Supplementary Material 1 for illustrative examples of multisectoral collaborations).“

VERSION 2 – REVIEW

REVIEWER	Shikako-Thomas, Keiko McGill University
REVIEW RETURNED	31-Aug-2023
GENERAL COMMENTS	Thanks for addressing all the comments and providing the Supplemental materials. I am sure these materials will contribute to expanding readers' understanding of the context and methods. I believe it in its current form the manuscript is ready for publication.